# Relationship between Grateful Disposition and Subjective Happiness of Korean Young Adults: Focused on Double Mediating Effect of Social Support and Positive Interpretation

**DOI:** 10.3390/bs13040287

**Published:** 2023-03-27

**Authors:** Jae-Sun An, Kyung-Hyun Suh

**Affiliations:** Department of Counseling Psychology, Sahmyook University, Seoul 01795, Republic of Korea

**Keywords:** gratitude, social support, positivity, happiness, young adults

## Abstract

This study aimed to identify the relationship between grateful disposition and the subjective happiness of young adults; it examined a sequential double mediating effect model of social support and positive interpretation on this relationship. The study participants included 389 male and female Korean young adults. The Korean version of Gratitude Questionnaire-6, a modified subscale of the SU Mental Health Test, Iverson et al.’s scale for social support, and the Subjective Happiness Scale were used. PROCESS Macro 3.5 Model 6 was used to analyze the double mediating effect. The correlation analysis showed that grateful disposition was positively correlated with social support, positive interpretation, and subjective happiness in young adults. Moreover, social support was positively correlated with positive interpretation and subjective happiness, whereas positive interpretation was positively correlated with subjective happiness. In addition, the sequential mediating effect of social support and positive interpretation on grateful disposition and the subjective happiness of young adults was significant. This study confirmed the determinant roles of social support and positive interpretation in grateful disposition and the subjective happiness of young adults, providing useful information for planning future studies and developing education materials and interventions for cultivating grateful disposition in childhood and promoting happiness in young adults.

## 1. Introduction

Although there is no consensus on the age of early adulthood, Levinson considered it to be up to age 40 and generally stable after age 32; however, it was explained as a stressful transitional stage in which young people pursue independence, enter the adult world, and decide how to live the rest of their lives [1,2]. After analyzing repeated cross-sectional survey data of more than 100,000 individuals from 38 countries, Hovi found that subjective well-being in later life was determined by people’s experiences in their early adulthood [3]. Recently, young adults in South Korea have experienced distress due to COVID-19, unemployment, and other economic difficulties [4]. Therefore, this study aimed to investigate the psychological variables related to subjective happiness in young adults.

The MZ generation—a combination of Millennials and Generation Z—is a term used in South Korea to refer to young adults, with focus on how they seek happiness, having different needs from those of the older generations [5]. However, this study focused on young adults’ inner psychological characteristics; that is, temperament or dispositional traits, which affect their happiness in young adulthood. In this study, gratitude was chosen as such a variable, and the tendency toward gratitude was regarded as a grateful disposition [6]. It was assumed that such an inner characteristic, grateful disposition, could affect an individual’s emotion or cognition, allowing them to experience happiness.

Watkins believed that gratitude—gratitude disposition or gratitude behavior—is related to subjective well-being, which represents happiness [7]. Many studies have empirically found that gratitude is correlated with subjective well-being [8,9,10]. A longitudinal study verified a structural equation model that revealed the relationship between gratitude disposition and life satisfaction, which is a factor in subjective well-being [11]. Watkins et al. explored the cause and effect relationship between gratitude and happiness based on various studies and theories and suggested that gratitude can improve happiness [12]. Therefore, in the field of positive psychology, attempts have been made to use gratitude as a clinical intervention to promote individuals’ well-being and happiness [13].

After verifying that gratitude is positively correlated with various factors of subjective well-being, Watkins et al. concluded that gratitude promotes people’s feelings of happiness through a positive mood [14]. They emphasized the affective role of gratitude, which affects individuals’ happiness. Additionally, they assumed that grateful thinking improves mood, which induces feelings of happiness. In fact, psychological treatments that utilize positive emotional effects for enhancing happiness do so by bringing about a change in thinking; that is, through cognitive intervention [15]. Therefore, this study focused on the cognitive component of gratitude—grateful thinking related to grateful disposition and happiness in young adults.

Burzynska-Tatjewska and Stolarski found that changing people’s perspectives on undesirable past events, which trigger negative emotions, by practicing gratitude can make them happier and improve their well-being [16]. In other words, people can be happier if they positively modify their interpretation of past experiences. Bryant introduced the concept of savoring belief, which includes thoughts of appreciation and gratitude for one’s life [17]. Bryant’s Saving Beliefs Inventory (SBI) measures a person’s beliefs about their past, present, and future lives; however, in this study, we focus on the positive interpretations of past and present lives.

Furthermore, it was assumed that a grateful disposition can change the social environment; that is, an individual’s grateful disposition can make their surrounding social environment more positive. Thus, it was assumed that people’s practice of gratitude will increase the likelihood of them receiving help from others around them. Previous studies have shown a significant correlation between gratitude and social support [18,19]. A longitudinal study found that people’s gratitude trait can lead to social support and assist in dealing with stress and depression [20]. Deichert et al. found that gratitude can enhance psychological well-being by boosting the beneficial effects of social support [21]. Furthermore, a study revealed that social support could mediate the relationship between gratitude and depression [22]. Based on previous studies, it can be assumed that people with a high gratitude disposition are likely to experience more happiness as a result of increased social support.

However, happiness can be both a cause and consequence of gratitude [12]; there are many positive things that can make people feel happy, which makes them feel grateful. Therefore, in this study, it was assumed that if grateful disposition is high, there is more social support, which results in people feeling happy. In addition, it was assumed that people can be happy by positively interpreting their lives, if they receive social support. Rash et al. found that grateful contemplation, as a psychological intervention, can enhance life satisfaction in the long term and that such an effect was moderated by the gratitude trait [23]. This implies that the gratitude trait is a dispositional factor that precedes gratitude practice and the consequences of gratitude.

## 2. Materials and Methods

### 2.1. Research Design and Hypothesis

It was hypothesized that grateful disposition directly increases young adults’ subjective happiness. The grateful disposition of young adults is positively, directly, and indirectly related to subjective happiness and social support. This hypothesis can be proposed because it has been empirically proven that social support is positively correlated with happiness [24,25]. Moreover, it is likely that young adults are more likely to positively interpret their lives if they have a high grateful disposition. Thus, in this study, an attempt was made to verify a model that includes a path from grateful disposition to subjective happiness through positive interpretation. As positivity or positive thoughts are important components of happiness, the rationale behind this path model is sound [26]. Moreover, the model includes a sequential double mediation effect of social support and positive interpretation between grateful disposition and the subjective happiness of young adults (Figure 1).

### 2.2. Participants

A total of 389 male and female South Korean young adults participated in this study. The data were collected via Embrain, an online survey company, from 10 August 2022 to 19 August 2022. They were young adults who had consented to participate in the study from among those listed in the online survey company. Using G*Power 3.1.9.7, the minimum sample size required to reach statistical significance with the number of predictors was 176, with significance level (0.05), power (0.95), and effect size (0.10). However, considering the differences in the predictors and criterion variables due to sex, the minimum sample size required was 352.

### 2.3. Participants’ Characteristics

Among the participants, 191 (49.1%) were male and 186 (50.9%) were female. Their ages ranged from 19–39 years, with a mean of 29.58 ± 5.48 years. As shown in Table 1, 205 (52.7%) participants were 19 or in their 20s and 194 (46.3%) in their 30s. A total of 279 (71.7%) participants were college graduates, 80 (20.6%) high school graduates, and 30 (7.7%) had completed graduate school courses or graduated graduate school.

Regarding the participants’ marital status, 290 (74.6%) were single, 97 (24.9%) married, and 2 (0.5%) divorced. Additionally, 298 (76.6%) participants were living with their family or others and 91 (23.4%) reported living alone. Furthermore, 271 (69.7%) participants reported having a religion, 118 (30.3%) reported not having a religion, 291 (74.8%) were employed, and 98 (25.2%) were unemployed.

### 2.4. Data Collection

The study was approved by the Institutional Review Board (IRB) of Sahmyook University, and all data collection procedures were conducted ethically. Before data were collected online, written informed consent was obtained from all participants. Moreover, they were informed that they could withdraw from the study at any time and, furthermore, if they felt psychologically uncomfortable during the survey, they could seek assistance to reduce their distress through debriefing. They were also assured that all data obtained anonymously will be used only for research purposes, stored in an encrypted computer for three years, and then discarded.

### 2.5. Instruments

#### 2.5.1. Gratitude Questionnaire-6

The Gratitude Questionnaire-6 (GQ-6) developed by McCullough et al. was used to measure participants’ gratitude disposition [6]. In this study, the Korean version of the scale validated by Kwon et al. (K-GQ-6) was used [27]. It comprises six items, such as “I have so much in life to be thankful for” and “I am grateful to a wide variety of people.” Two items were reversed scored; all items were rated on a seven-point Likert scale ranging from 1 (*strongly disagree*) to 7 (*strongly agree*). In a validation study of the K-GQ-6, the internal consistency (Cronbach’s α) of the items was 0.85 [27], and in this study it was 0.91.

#### 2.5.2. Positive Interpretation Questionnaire

To measure the tendency to positively interpret life events in early adulthood, this study used modified items from the positivity subscale for the protective factors of mental health in the SU Mental Health Test developed by Suh et al. [28]. This questionnaire comprises five items with a single factor; examples of items are “I get so many things from my life” and “Looking back on my life so far, there are many positive things.” The items were rated on a six-point Likert scale, ranging from 1 (*strongly disagree*) to 6 (*strongly agree*). In the scale development study, the internal consistency (Cronbach’s α) of the items was 0.90, and the test–retest reliability was 0.70 [28]; in this study, the Cronbach’s α was 0.92.

#### 2.5.3. Scale for Social Support

The social support that participants received from others was measured using items from the scale developed by Iverson et al.; this scale was translated and validated for Koreans by Nos [29,30]. This scale measures how individuals perceive themselves as being socially supported by others. Specifically, this scale comprising four items measures the extent to which an individual perceives social support from their superiors (seniors), colleagues (friends), and family, and whether they receive help in problem solving and emotional support. In this study, the total score was used in the analysis; the higher the score, the more perceived support from superiors, peers, and family. The items were rated on a five-point Likert scale ranging from 1 (*never*) to 5 (*very often*). The internal consistency (Cronbach’s α) of the items in No’s study was 0.89 [30], and in this study it was 0.91.

#### 2.5.4. Subjective Happiness Scale

Happiness experienced in early adulthood was measured using the Subjective Happiness Scale (SHS) developed by Lyubomirsky and Lepper [31]. In this study, the scale translated by Kim [32] was used. This scale comprises four items rated on seven points related to the state of happiness; however, one item that inquires about the state of unhappiness is reverse scored. In this study, the internal consistency (Cronbach’s α) of the items was 0.89.

### 2.6. Statistical Analysis

Data were analyzed using IBM SPSS (Statistical Package for Social Sciences) Statistics for Windows 23.0 and PROCESS Macro 3.5. The skewness and kurtosis of the variables were calculated to determine whether the criteria for parametric statistical analysis was met. Pearson’s product moment correlational analysis was performed using SPSS, and a sequential double moderating mediating effect was examined using PROCESS Macro 3.5 Model 6 [33]. The significance of the mediating model was also analyzed using 5000 bootstrap replicates with a 95% confidence interval. Because none of the demographic profiles collected met the criteria for a confounding variable, the model was not adjusted with covariates.

## 3. Results

### 3.1. Relationship between Variables Involved in Subjective Happiness of Young Adults

Table 2 presents the results of the correlational analysis of grateful disposition, social support, positive interpretation, and the subjective happiness of Korean young adults. None of the absolute values of skewness and kurtosis exceeded 1, showing that the variances of all variables in this study were close to the normal distribution, satisfying the condition to perform parametric statistical analyses [34].

The correlational analysis revealed that grateful disposition was positively correlated with social support (*r* = 0.690, *p* < 0.001), positive interpretation (*r* = 0.767, *p* < 0.001), and subjective happiness (*r* = 0.673, *p* < 0.001). Social support was also positively correlated with positive interpretation (*r* = 0.754, *p* < 0.001) and subjective happiness (*r* = 0.601, *p* < 0.001), whereas positive interpretation was positively correlated with subjective happiness (*r* = 0.781, *p* < 0.001).

### 3.2. Verification of the Double Mediation Model for the Subjective Happiness of Young Adults

This study examined the sequential double mediating effect of social support and positive interpretation on grateful disposition and subjective happiness in Korean young adults (Table 3). Before conducting this analysis, a check was conducted for statistical multicollinearity, because grateful disposition, social support, and positive interpretation were closely correlated. The statistical multicollinearity problem occurs if the tolerance is less than 0.2 or 0.1 and the variance inflation factors (VIFs) are higher than 5 or 10 [35]. The tolerances of predictors were 0.315–0.401 and VIFs were 2.493–3.177, indicating no multicollinearity problem. Moreover, the Durbin–Watson statistic was 2.059, which implies that no autocorrelation was detected in the sample, as the value was close to 2.

The results revealed that grateful disposition positively influenced social support (*B* = 0.609, *p* < 0.001) and significantly influenced positive interpretation (*B* = 0.354, *p* < 0.001) and the subjective happiness (*B* = 0.137, *p* < 0.001) of young adults in this model. In addition, social support significantly influenced positive interpretation (*B* = 0.365, *p* < 0.001), but not subjective happiness (*B* = −0.015, *p* = 0.713) in this model. Moreover, positive interpretation positively influenced the subjective happiness of young adults (*B* = 0.647, *p* < 0.001).

Figure 2 shows that grateful disposition alone significantly influenced the subjective happiness of young adults (*β* = 0.673, *p* < 0.001). Although it continued to significantly influence subjective happiness (*β* = 0.184, *p* < 0.001), its coefficient decreased when social support and positive interpretation were added as mediating variables. This implies that social support and positive interpretation partially sequentially mediated grateful disposition and the subjective happiness of young adults in this model.

Using 95% confidence intervals from 5000 bootstrap replications, the double mediating effect of social support and positive interpretation in the relationship between grateful disposition and the subjective happiness of young adults was examined (Table 4).

The total mediating effect was 0.364 (0.2870–0.4475), which was significant, as no zero existed between the upper and lower bounds of bootstrapping at 95% confidence intervals. Examining the simple mediating effect showed that the path from grateful disposition to subjective happiness through social support was not significant (−0.0693–0.0508). However, the path from grateful disposition to subjective happiness through positive interpretation was significant (0.1616–0.3072). Furthermore, the sequential double mediating effect of social support and positive interpretation on grateful disposition and subjective happiness (grateful disposition → social support → positive interpretation → subjective happiness) was 0.144 (0.1069–0.1838), which was also significant. 

Furthermore, an analysis was conducted to determine whether there were any differences in the indirect effect sizes examined in this study (Table 4). First, the effect sizes of both indirect paths (A → C → D and A → B → C → D) were greater than that of the first indirect path (A → B → D). In addition, the effect size of positive interpretation on grateful disposition and subjective happiness (A → C → D) was significantly greater than that of the sequential double mediating effect of social support and positive interpretation (A → B → C → D) (0.0089–0.1745).

## 4. Discussion

The present study investigated the relationships between grateful disposition, social support, positive interpretation, and the subjective happiness of Korean young adults. Furthermore, it examined the double mediating effect of social support and positive interpretation on grateful disposition and the subjective happiness of young adults. The implications of this study’s findings are discussed below.

As hypothesized in this study, grateful disposition and social support were positively correlated in young adults. The grateful disposition shared approximately 47.6% of the variance (*r* = 0.690) with social support in young adults. According to Wood et al., if gratitude can directly foster social support, it can prevent individuals from experiencing stress and depression [20]; thus, this finding has clinical significance. Additionally, this indicates that gratitude is not merely a religious or moral affect [36]. This study revealed that grateful disposition or gratitude may create a social environment for maintaining mental health or promoting well-being and happiness.

The grateful disposition was closely correlated with a positive interpretation of life in young adults. The grateful disposition shared approximately 58.8% of the variance (*r* = 0.767) with positive interpretation in young adults. This result indicates that the higher the grateful disposition of young adults, the more likely they are to interpret life positively. In the model analyzed in this study, grateful disposition had a considerable influence on positive interpretation, except for the influence of social support. This result supports the view that thinking positively is related to practicing gratitude, without support from others and when undesirable things happen [16]. This viewpoint is also supported by the finding that the path from gratitude disposition to subjective happiness in young adults through positive interpretation was significant in the mediating model. 

In this study, social support was closely correlated with the subjective happiness of young adults. It accounted for approximately 36.1% (*r* = 0.601) of the happiness in young adults. However, in the mediation model, the direct effect of social support was not significant. This result suggests that even if young adults receive social support, they do not necessarily experience happiness unless they interpret it positively. Furthermore, this implies that happiness does not originate from the outside, but from the inside. Naturally, when people receive social support or when good things happen, they think more positively and are more likely to be happy; however, this study shows that happiness is not a direct result of social support, nor is it implied when good things happen. Using gratitude as a clinical intervention to promote positive emotions and happiness, Sheldon and Lyubomirsky succeeded in making people think more positively about themselves [37]. This study also suggests that positive thinking as a cognitive strategy is essential for enhancing happiness.

Notably, the sequential mediating path from grateful disposition to subjective happiness through social support and positive interpretation was significant. The effect of grateful disposition on subjective happiness was greatly reduced when social support and positive interpretation were entered as mediating variables. This suggests that social support and positive interpretation play determinant roles in the relationship between grateful disposition and subjective happiness. Although the sequential double mediating effect of social support and positive interpretation was smaller than the mediating effect of positive interpretation on grateful disposition and subjective happiness, it was noteworthy, both clinically and educationally. Similar to previous studies, this study suggests that cognitive interventions that allow people to positively interpret their life events in young adulthood can be effective in promoting individual mental health and well-being [13,15]. However, as this study provides valuable information for maintaining mental health and promoting subjective well-being, the relationship between the study variables needs to be reconfirmed and the cause and effect relationships must be proven in future studies. 

However, this study shows that by practicing gratitude, young adults can receive more social support, which can foster positive thoughts about their own lives; that is, grateful beliefs or savoring beliefs, making them happier. Regarding the clinical and educational implications of these findings, it is important to make people aware that if they are grateful, they are more likely to receive help or support from others. It is natural to be grateful for help and support from others; however, teaching children that by practicing gratitude they will receive more help and support from others is important. Consequently, children will develop a grateful disposition in childhood, which can have long-term effects on their mental health and well-being.

This study has some limitations that must be considered when interpreting the results. First, the study sample of young adults who were registered by an online survey research company cannot be considered representative of all Korean young adults. In particular, many of the participants graduated from universities and those with a relatively high educational qualification level responded to the online survey. Therefore, it is necessary to confirm the study results using other sampling methods. Second, although this study hypothesized and discussed the cause and effect relationship between the variables based on previous studies and logic, causation cannot be concluded with certainty based on the results of a correlational study; it should be interpreted carefully and an experimental study design is required to do so. Finally, the results of the self-report psychological test inevitably included errors and response bias. 

## 5. Conclusions

This study found a determinant role of social support and positive interpretation in the relationship between grateful disposition and the subjective happiness of Korean young adults. This study suggests that positive interpretation is a key element of grateful disposition and an important factor in helping young adults experience happiness. In addition, this study revealed that grateful disposition in young adults can lead to social support from others, thereby making them feel happy by thinking positively about their lives. Therefore, despite these limitations, this study contributes academically to planning future studies, educationally by emphasizing the importance of developing grateful disposition in childhood, and clinically for preparing interventions to promote subjective well-being and happiness in young adults.

## Figures and Tables

**Figure 1 behavsci-13-00287-f001:**
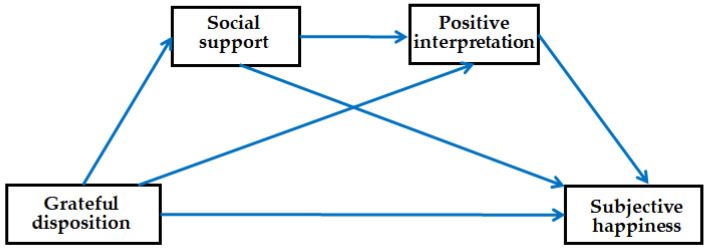
Double sequential mediation model.

**Figure 2 behavsci-13-00287-f002:**
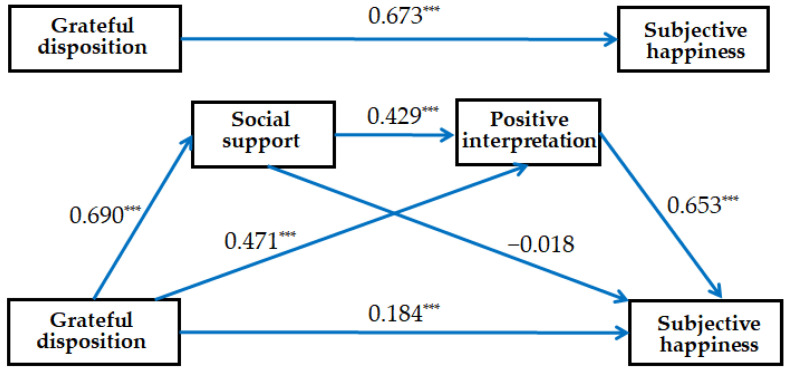
Examined double mediation model of social support and positive interpretation on grateful disposition and subjective happiness (*β*, standardized coefficients; *** *p* < 0.001).

**Table 1 behavsci-13-00287-t001:** Participants’ characteristics (*N* = 389).

Variables		Frequency	Percent (%)
**Gender**	Males Femals	191 198	49.1 50.9
**Age group**	19 or 20s 30s	205 194	52.7 47.3
**Educational qualification**	High school College Graduate school	80 279 30	20.6 71.7 7.7
**Marital status**	Single Married Divorced	290 97 2	74.6 24.9 0.5
**Residence type**	Living alone Living with other(s)	91 298	23.4 76.6
**Religion**	Having a religion Having no religion	271 118	69.7 30.3
**Occupation**	Employed Unemployed	291 98	74.8 25.2

**Table 2 behavsci-13-00287-t002:** Correlational matrix of grateful disposition, social support, positive interpretation, and subjective happiness in Korean young adults (*N* = 389).

Variables	1	2	3	4
**1. Grateful disposition**	1			
**2. Social support**	0.690 ***	1		
**3. Positive interpretation**	0.767 ***	0.754 ***	1	
**4. Subjective happiness**	0.673 ***	0.601 ***	0.781 ***	1
** *M* **	28.98	27.87	20.86	17.48
** *SD* **	6.95	6.15	5.23	5.18
**Skewness**	−0.10	−0.08	−0.29	−0.28
**Kurtosis**	−0.32	−0.25	−0.29	−0.26

*** *p* < 0.001.

**Table 3 behavsci-13-00287-t003:** Double mediating effect of social support and positive interpretation on grateful disposition and subjective happiness.

Variables	*B*	*S.E.*	*t*	LLCI	ULCI
**Mediating Variable Model (Outcome Variable: Social support)**					
Constant	10.212	0.969	10.53 ***	8.3061	12.1181
Grateful disposition	0.609	0.033	18.73 ***	0.5455	0.6734
**Mediating Variable Model (Outcome Variable: Positive interpretation)**					
Constant	0.417	0.726	0.58	−1.0098	1.8438
Grateful disposition	0.354	0.030	11.94 ***	0.2958	0.4124
Social support	0.365	0.034	10.89 ***	0.2994	0.4313
**Dependent Variable Model (Outcome Variable: Subjective happiness)**					
Constant	0.433	0.788	0.55	−1.1165	1.9828
Grateful disposition	0.137	0.038	3.64 ***	0.0629	0.2110
Social support	−0.015	0.042	−0.37	−0.0972	0.0666
Positive interpretation	0.647	0.055	11.71 ***	0.5385	0.7558

*** *p* < 0.001. LLCI: lower level for confidence interval; ULCI: upper level for confidence interval.

**Table 4 behavsci-13-00287-t004:** Indirect effects of the mediation model.

Path	Effect	*S.E*	BC 95% CI
Total indirect effect	0.364	0.041	0.2870~0.4475
Ind 1: A → B → C	−0.009	0.031	−0.0693~0.0508
Ind 2: A → C → D	0.229	0.037	0.1616~0.3072
Ind 3: A → B → C → D	0.144	0.020	0.1069~0.1838
Ind 1/Ind 2	−0.239	0.055	−0.3467~−0.1299
Ind 1/Ind 3	−0.153	0.040	−0.2322~−0.0753
Ind 2/Ind 3	0.085	0.042	0.0089~0.1745

A = Grateful disposition, B = Social support, C = Positive interpretation, D = Subjective happiness.

## Data Availability

The datasets analyzed in this study are available from the corresponding author upon reasonable request.

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
