# Peer review of "Relationship between Grateful Disposition and Subjective Happiness of Korean Young Adults: Focused on Double Mediating Effect of Social Support and Positive Interpretation"

_behavsci, 2023, doi:10.3390/bs13040287_

Round 1

Reviewer 1 Report

This study introduce diverse conclusions allowing to understand more clearly the relasionship in any community.  In my perspective it is important that this study shows that, by praticing gratitude, young adults can receive more social support. other important result is that children will developba grateful disposition in chioldhood, which can have long-term effects on their mental healt and well- being. Thank you ! 

Author Response

Thank you very much for your comments to improve the quality of this articles and we learned a lot.

The revised parts were marked in red, and we included the page and line of the revised part.

Response to Reviewer 1 Comments

Point 1. This study introduce diverse conclusions allowing to understand more clearly the relationship in any community.  In my perspective it is important that this study shows that, by practicing gratitude, young adults can receive more social support. other important result is that children will develop grateful disposition in childhood, which can have long-term effects on their mental health and well- being.

Response 1: Thank you very much for the good evaluation. Based on the comments of other reviewers, it was further supplemented and carefully revised.

Reviewer 2 Report

I congratulate them for sharing their study with the scientific community and addressing a topic that can be of great interest to promoting well-being and happiness. In my opinion, your work is very complete, but susceptible to improvement, as any human task. Therefore, please consider my comments and suggestions as a sign of interest in your work, and desire to make it more accurate.

Title and abstract

The title of the article gives the reader an idea of its content, and the abstract provides an informative summary of what was done and the results found. However, it is advisable to include the design used in the title or abstract, as it quickly orients the potential reader to the scope of the results and facilitates a more accurate classification of the work in bibliographic databases.

Introduction

The objective of the study and its hypotheses are sufficiently justified, specifically formulated and clearly and precisely stated. The study analyses the relationships between gratitude disposition, social support, positive interpretation, and subjective happiness of Korean young adults. In addition, it examines the dual mediating effect of social support and positive interpretation on young adults' gratitude disposition and subjective happiness.

Materials and Methods.

The authors describe in detail the composition and characteristics of the sample and, roughly, the sample environment. In addition, they clearly define the study variables and the instruments used to measure them, as well as how the data were collected and the statistical analyses of the data were performed.

However, we believe that this section would be more complete if the authors included the study design, although from reading the method it is easy to deduce that this is a cross-sectional study. 

On the other hand, we believe that it would also be good to indicate the relevant dates of the study, including the recruitment and data collection periods. Furthermore, although the work of choosing and selecting the participants was carried out by a company (Embrain), it would be useful for the authors to explain the selection criteria they provided to the company.

Finally, we believe it would be good for them to describe, in the section devoted to statistical analyses, the efforts made to address possible sources of bias.

Results

The presentation of the results is accurate, very close to the proposed objectives and hypotheses, and complete. Therefore, we have nothing to add in this regard.

Discussion and conclusions

The authors summarize the main results in relation to the objectives of the study. They also comment on the limitations of the study: the sample is not representative of the population of interest, the study design itself does not allow causal relationships to be established, although the cause-effect relationship between the variables studied was hypothesized. In addition, the data were obtained with self-report instruments, so response bias was inevitable.

Consequently, the authors are cautious in interpreting the results, and although these are not generalizable, they may contribute to the planning of future intervention studies that promote subjective well-being and happiness in young adults.

Author Response

Thank you very much for your comments to improve the quality of this articles and we learned a lot.

The revised parts were marked in red, and we included the page and line of the revised part.

Response to Reviewer 2 Comments

Point 1. (Title and abstract) The title of the article gives the reader an idea of its content, and the abstract provides an informative summary of what was done and the results found. However, it is advisable to include the design used in the title or abstract, as it quickly orients the potential reader to the scope of the results and facilitates a more accurate classification of the work in bibliographic databases...

Response 1: Thank you for your nice comment. As you advised, the title has been modified to reveal more of the research design as shown below. In addition, the research purpose of the abstract was slightly modified to reveal the research design. (Line 3-4, 9-11)

Relationship between Grateful Disposition and Subjective Happiness of Korean Young Adults: Focused on Double Mediating Effect of Social Support and Positive Interpretation

This study aimed to identify the relationship between grateful disposition and subjective happiness of young adults, and examined a sequential double mediating effect model of social support and positive interpretation on this relationship.

Point 2. (Introduction) The objective of the study and its hypotheses are sufficiently justified, specifically formulated and clearly and precisely stated. The study analyses the relationships between gratitude disposition, social support, positive interpretation, and subjective happiness of Korean young adults. In addition, it examines the dual mediating effect of social support and positive interpretation on young adults' gratitude disposition and subjective happiness.

Response 2: Thank you for the good evaluation.

Point 3. (Materials and Methods) We believe that this section would be more complete if the authors included the study design, although from reading the method it is easy to deduce that this is a cross-sectional study.

Response 3: Thank you very much for your good comment on this part. As you advised, the research design and hypothesis are included in the method section as follows: (Line 93-105)

2.1. Research design and hypothesis

It was hypothesized that grateful disposition directly increases young adults’ subjective happiness. The grateful disposition of young adults is positively, directly, and indirectly related to subjective happiness and social support. This hypothesis can be proposed because it has been empirically proven that social support is positively correlated with happiness [24,25]. Moreover, it is likely that young adults are more likely to positively interpret their lives if they have a high grateful disposition. Thus, in this study, it was attempted to verify a model that includes a path from grateful disposition to subjective happiness through positive interpretation. As positivity or positive thoughts are important components of happiness, the rationale behind this path model is sound [26]. Moreover, the model includes a sequential double mediation effect of social support and positive interpretation between grateful disposition and subjective happiness of young adults (Figure 1).

Point 4. (Materials and Methods) On the other hand, we believe that it would also be good to indicate the relevant dates of the study, including the recruitment and data collection periods. Furthermore, although the work of choosing and selecting the participants was carried out by a company (Embrain), it would be useful for the authors to explain the selection criteria they provided to the company.

Response 4: Thank you for your valuable comment. As you pointed out, it was described below: (Line 112-114)

The data were collected via Embrain, an online survey company, from August 10, 2022, to August 19, 2022. They were young adults who had consented to participate in the study from among those listed in the online survey company.

Point 5. (Materials and Methods) We believe it would be good for them to describe, in the section devoted to statistical analyses, the efforts made to address possible sources of bias.

Response 5: Thank you for your comment. As you advised, it was included in statistical analysis section as follows: (Line 191-193)

Because none of the demographic profiles collected met the criteria for confounding variable, the model was not adjusted with covariates.

Point 6. (Results) The presentation of the results is accurate, very close to the proposed objectives and hypotheses, and complete. Therefore, we have nothing to add in this regard.

Response 6: Thank you for the good evaluation.

Point 7. (Discussion and conclusions) The authors summarize the main results in relation to the objectives of the study. They also comment on the limitations of the study: the sample is not representative of the population of interest, the study design itself does not allow causal relationships to be established, although the cause-effect relationship between the variables studied was hypothesized. In addition, the data were obtained with self-report instruments, so response bias was inevitable. Consequently, the authors are cautious in interpreting the results, and although these are not generalizable, they may contribute to the planning of future intervention studies that promote subjective well-being and happiness in young adults.

Response 7: We agree with your comments. So, the limitations of the study regarding this concern include the following.  (Line 329-333)

Second, although it was hypothesized and discussed the cause-and-effect relationship between the variables based on previous studies and logic, causation cannot be concluded with certainty based on the results of a correlational study; it should be interpreted carefully and an experimental study design is required to do so.

Reviewer 3 Report

The paper has got a very suggestive title and a promissing idea about a very relevant topic for social psychology such as grateful disposition and subjective happiness. I have some questions, comments and suggestions for the authors:

Abstract: there is no mention to the aims/hypotheses/methods in the abstract, only about the analysis (correlation + PROCESS Macro 3.5 model). It would be proper to include one sentence describing briefly the methodology of the study

Introduction: The reference to (Levinson, 1977) is a classic, but maybe it would be better to mention more actual authors.

In Introduction,  the reasons of the study are indicated in first person (We). Suggestion to use third person in redaction.

The description of the hypotheses and the measures  can be better included in the Methods part of the paper.

Methods:  There is a clear bias in the selection of the sample, most of them are students. Have you done any kind of analysis to decrease/control this bias?

There is a lack of information in 2.1 Participants ("The data were collected via Embrain, an online survey company. "). I totally understand the data protection and confidenciality about the study, but it´s necessary more information about how had you controlled the sample selection in order to clarify from internal validity to external validity of the study.

Results: 

Results are wonderful and very well presented, thanks a lot. Anyway, data in text are same than data in table, so it´would be more feasible for the reader to find in the text only significative data.

Discusion.

To support your discussion, it´s necessary to make reference to more recient studies (the first part of discussion is empty of references).

Limitations should be  moved at the end of Discussion section.

Author Response

Thank you very much for your comments to improve the quality of this articles and we learned a lot.

The revised parts were marked in red, and we included the page and line of the revised part.

Response to Reviewer 3 Comments

Point 1. (Abstract) there is no mention to the aims/hypotheses/methods in the abstract, only about the analysis (correlation + PROCESS Macro 3.5 model). It would be proper to include one sentence describing briefly the methodology of the study.

Response 1: Thank you for your valuable comment. As you advised, the research purpose in abstract was slightly modified to reveal the research design, and instruments used in this study were included as below: (Line 9-14)

This study aimed to identify the relationship between grateful disposition and subjective happiness of young adults, and examined a sequential double mediating effect model of social support and positive interpretation on this relationship. ….. The Korean version of Gratitude Questionnaire-6, a modified subscale of SU Mental Health Test, Iverson et al.’s scale for social support, and the Subjective Happiness Scale were used.

Point 2. (Introduction) The reference to (Levinson, 1977) is a classic, but maybe it would be better to mention more actual authors.

Response 2: Thank you for your advice. In addition to Levinson, we have added one more reference as below. (Line 28-30, 369-370)

Although there is no consensus on the age of early adulthood, Levinson considered it to be up to age 40, generally stable after age 32; however, it was explained as a stressful transitional stage in which young people pursue independence, enter the adult world, and decide how to live the rest of their lives [1, 2].

Matud, M.P.; Díaz, A.; Bethencourt, J.M.; Ibáñez, I. Stress and psychological distress in emerging adulthood: A gender analysis. J. Clin. Med. 2020, 9, 2859. https://doi.org/10.3390/jcm9092859.

Point 3. (Introduction) In Introduction, the reasons of the study are indicated in first person (We). Suggestion to use third person in redaction.

Response 3: Thank you very much for your good comment. All sentences that were expressed as first person (We) have been modified to third person as shown below: (whole manuscript)

Furthermore, it was assumed that a grateful disposition can change the social environment….

Point 4. (Introduction) The description of the hypotheses and the measures can be better included in the Methods part of the paper.

Response 4: As you advised, the hypothesis are included in the method section as follows: (Line 93-105)

2.1. Research design and hypothesis

It was hypothesized that grateful disposition directly increases young adults’ subjective happiness. The grateful disposition of young adults is positively, directly, and indirectly related to subjective happiness and social support. This hypothesis can be proposed because it has been empirically proven that social support is positively correlated with happiness [24,25]. Moreover, it is likely that young adults are more likely to positively interpret their lives if they have a high grateful disposition. Thus, in this study, it was attempted to verify a model that includes a path from grateful disposition to subjective happiness through positive interpretation. As positivity or positive thoughts are important components of happiness, the rationale behind this path model is sound [26]. Moreover, the model includes a sequential double mediation effect of social support and positive interpretation between grateful disposition and subjective happiness of young adults (Figure 1).

Point 5. (Methods) There is a clear bias in the selection of the sample, most of them are students. Have you done any kind of analysis to decrease/control this bias?

Response 5: Thank you for your comment. Not many college students participated in this study. Many of the participants had college degrees. Even so, this point is also a limitation of the study, so it is included in the manuscript as follows. (Line 325-328)

First, the study sample of young adults who were registered by an online survey research company cannot be considered representative of all Korean young adults. In particular, many of the participants graduated from universities, and those with a relatively high educational qualification level responded to the online survey. .

Point 6. (Methods). There is a lack of information in 2.1 Participants ("The data were collected via Embrain, an online survey company. "). I totally understand the data protection and confidenciality about the study, but it´s necessary more information about how had you controlled the sample selection in order to clarify from internal validity to external validity of the study.

Response 6: We included what you advised as below. (Line 112-114)

The data were collected via Embrain, an online survey company, from August 10, 2022, to August 19, 2022. They were young adults who had consented to participate in the study from among those listed in the online survey company.

Point 7. (Results) Results are wonderful and very well presented, thanks a lot. Anyway, data in text are same than data in table, so it´would be more feasible for the reader to find in the text only significative data.

Response 7: Thank you for your advice. The table is presented in addition to the contents of the main text, but as you said, almost all of the contents in the table were included in the main text. I also summarized the contents in the table or described only the important contents in other papers, but it is difficult to find anything to exclude from the contents described here. It seems to be fine as it is because there is not much content, but if you specify something, we will exclude it.

Point 8. (Discussion). To support your discussion, it´s necessary to make reference to more recent studies (the first part of discussion is empty of references).

Response 8: We would appreciate your understanding that the first paragraph of the discussion is a summary of how this study was conducted and does not require references. Reference that is too old replaced with more recent one as shown below. (Line 275, 442)

Carr, D. Is gratitude a moral virtue? Philos. Stud. 2015, 172, 1475–1484. https://doi.org/10.1007/s11098-014-0360-6

Point 9. (Discussion) Limitations should be moved at the end of Discussion section..

Response 4: Thank you for your comment. As you advised, I have moved limitations to the end of the discussion section. (Line 324-334)

This study has some limitations that must be considered when interpreting the results. First, the study sample of young adults who were registered by an online survey research company cannot be considered representative of all Korean young adults. In particular, many of the participants graduated from universities, and those with a relatively high educational qualification level responded to the online survey. Therefore, it is necessary to confirm the study results using other sampling methods. Second, although it was hypothesized and discussed the cause-and-effect relationship between the variables based on previous studies and logic, causation cannot be concluded with certainty based on the results of a correlational study; it should be interpreted carefully and an experimental study design is required to do so. Finally, the results of the self-report psychological test inevitably included errors and response bias.

  1. Conclusions
